



# Atmospheric rotating rig testing of a swept blade tip and comparison with multi-fidelity aeroelastic simulations

Thanasis Barlas, Georg Raimund Pirrung, Néstor Ramos-García, Sergio González Horcas, Ang Li, and Helge Aagaard Madsen

DTU Wind Energy, Frederiksborgvej 399, 4000 Roskilde, Denmark

**Correspondence:** Thanasis Barlas (tkba@dtu.dk)

**Abstract.**

One promising design solution for increasing the energy production of modern horizontal axis wind turbines is the installation of curved tip extensions. However, since the aeroelastic response of such geometrical add-ons has not been characterized yet, there are currently uncertainties in the application of traditional aerodynamic numerical models. The objective of the present work is twofold. On the one hand, it represents the first effort in the experimental characterization of curved tip extensions in atmospheric flow. On the other hand, it includes a comprehensive validation exercise, accounting for different numerical models for aerodynamic loads prediction. The experiments consist of controlled field tests in the outdoor rotating rig at the Risø campus of the Technical University of Denmark (DTU), and consider a swept tip shape. This geometry is the result of an optimized design, focusing on locally maximizing power performance within load constraints compared to an optimal straight tip. The tip model is instrumented with spanwise bands of pressure sensors and is tested in atmospheric inflow conditions. A range of fidelity of aerodynamic models is then utilized to aeroelastically simulate the test cases and to compare with the measurement data. Those aerodynamic codes include a blade element momentum (BEM) method, a vortex-based method coupling a near-wake model with a far-wake model (NW), a lifting-line hybrid wake model (LL) and fully resolved Navier-Stokes computational fluid dynamics (CFD) simulations. Results show that the measured mean normal loading can be captured well with the vortex-based codes and the CFD solver. The observed trends in mean loading are in good agreement with previous wind tunnel tests of a scaled and stiff model of the tip extension. The CFD solution shows a highly three-dimensional flow at the very outboard part of the curved tip that leads to large changes of the angle of the resultant force with respect to the chord. Turbulent simulations using the BEM code and the vortex codes resulted in a good match with the measured standard deviation of the normal force, with some deviations of the BEM results due to the missing root vortex effect.



## 1 Introduction

The trend of reducing the levelized cost of energy (LCOE) of horizontal axis wind turbines through increasing rotor size has long been established. To achieve this, the challenges of scaling must be overcome through innovative turbine design and control strategies (Veers et al., 2019). One promising blade design concept is advanced aeroelastically optimized blade tip
extensions, which could drive rotor upscaling in a modular and cost-effective way.

Existing bibliography relevant to wind turbine applications typically focuses on winglets and aerodynamic tip shapes, with limited testing of generalized curved shapes in controlled or atmospheric conditions (Johansen and Sørensen, 2006; Gaunaa and Jeppe, 2007; Gertz et al., 2012; Hansen and Mühle, 2018; Sessarego et al., 2020).

Previous related work by the authors focused on the aeroelastic optimization of curved tip extensions (Barlas et al., 2021a)
and wind tunnel testing (Barlas et al., 2021b). In the present work, the aeroelastic response of a swept tip extension is investigated for application to horizontal axis wind turbines. Controlled field testing is performed, using the outdoor rotating test rig (RTR) at the Technical University of Denmark (DTU). The swept tip shape in focus is the result of a design optimization, focusing on locally maximizing power performance within load constraints compared to an optimal straight tip, for testing in the RTR. The tip model is instrumented with spanwise bands of pressure sensors and is tested in atmospheric inflow conditions.
A range of fidelity of aerodynamic models is utilized to aeroelastically simulate the test cases and is compared with the measurement data, namely a blade element momentum (BEM) model, a vortex-based method coupling a near-wake model with a far-wake model (NW), a lifting-line hybrid wake model (LL), and fully resolved Navier-Stokes simulations (CFD).

## 2 Tip model design

The tip shape presented in this work is an aeroelastically optimized tip which is mounted on DTU's rotating test rig (RTR)
(Madsen et al., 2015; Ai et al., 2019), whereas a scaled stiff version of it has been tested in the wind tunnel (Barlas et al., 2021b). The design optimization method used is described in (Barlas et al., 2021a) for a tip extension on a full scale wind turbine. The method of optimizing the tip for the RTR is essentially the same, while the baseline geometry and load envelope is defined by a reference straight tip, designed for optimal BEM performance on a three-bladed rotor (Table 1). Additionally, the structural sectional layup of the tip is parametrized using the software BECAS (Blasques et al., 2015). The reference tip
is designed using the FFA-W3-211 airfoil with fully turbulent wind tunnel polars (Bertagnolio et al., 2001) for a Reynolds number of $1.78 \times 10^6$, with a predefined length of 3 m (practical design constraint for testing on the outdoor rotating rig), mounted on the 8 m cylindrical boom of the RTR. The cylindrical sections of the boom are modelled with a drag coefficient of 0.8 and zero lift. The chord and twist distributions of the straight tip were determined from BEM performance for optimal power coefficient in operation at 30 rpm with $6\,\mathrm{m\,s^{-1}}$ inflow wind speed. The resulting aeroelastically optimized tip utilizing
sweep (Fig. 1), achieved a 19.58% increase in power with the baseline ultimate flapwise bending moment at the boom root and tip connection, when evaluated at an extreme turbulence case (class III-C) at $6\,\mathrm{m\,s^{-1}}$ in the aeroelastic code HAWC2 (Larsen and Hansen, 2007) using the near wake (NW) model (Madsen and Rasmussen, 2004; Pirrung et al., 2016, 2017a, b; Li et al., 2022). The Pareto front of the design optimization solutions is shown in Fig. 2. The design features a highly swept




**Table 1.** Design variables range and optimized values.

| variable | length [m] | chord [%] | twist [deg] | dihedral [deg] | sweep [deg] | cap width [%] | cap thickness [mm] | web location [%] |
|---|---|---|---|---|---|---|---|---|
| min | 3 | 40 | -5 | -5 | 0 | 10 | 2.3 | 70 |
| max | 3.6 | 100 | 5 | 0 | 30 | 20 | 9.2 | 80 |
| opt | 3.48 | 41 | -3.74 | -0.05 | 25.28 | 16 | 5.5 | 70 |

(in-plane offset) centerline (Fig. 3), slender chord distribution, and negative twist (to feather) distribution (Fig. 4) compared
to the reference straight tip, whereas the airfoil sections are aligned perpendicular to the centerline. The very last tapering of
the tip chord is only relevant for the fully resolved Navier-Stokes modeling, whereas the optimized chord distribution from the
NW model stops at a finite chord of 0.41 m at the 12 m span location. The optimized mass and flapwise stiffness distribution
as well as the structural layup are shown in Fig. 5 and Fig. 6.

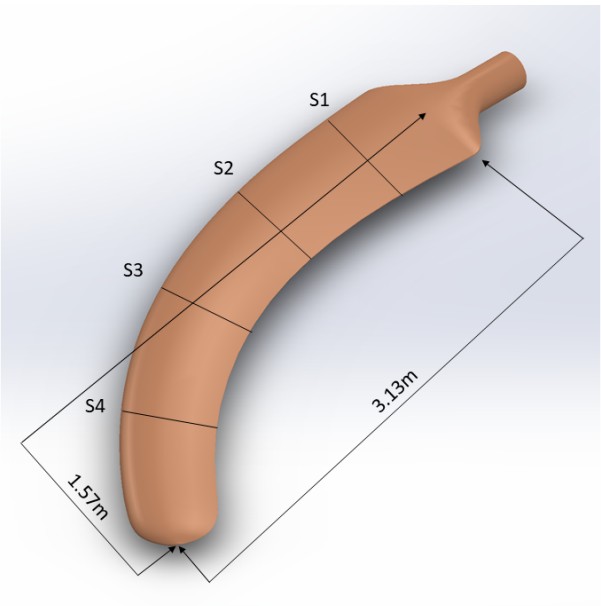

**Figure 1.** 3-D geometry of the tip, including basic dimensions and measurement sections.





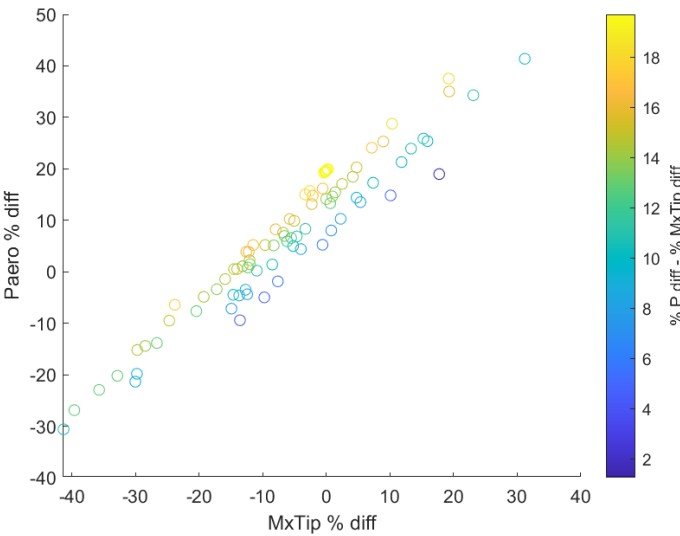

**Figure 2.** Pareto front of the optimization objectives.

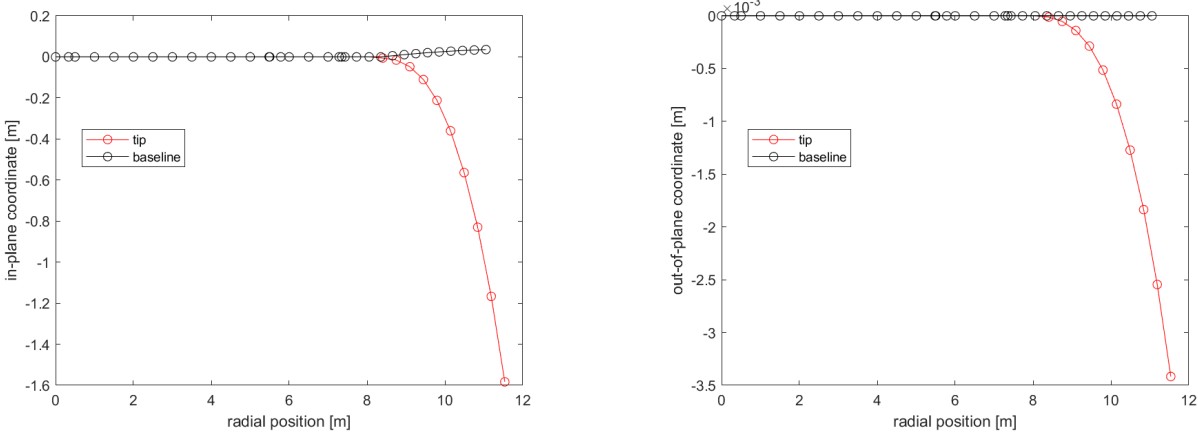

**Figure 3.** In-plane and out-of-plane coordinate of the centerline of the optimized tip design compared to the reference.

## 3 Rotating rig test setup

In order to fill in the gap between full-scale MW experiments and wind tunnel tests, the 95 kW Tellus turbine (Madsen and Petersen, 1990) situated at the test field at DTU Risø campus plays an important role as a test bed for aerodynamic and aero-servo-elastic experiments (Madsen et al., 2015; Ai et al., 2019). The 5° tilted rotor on the turbine has been replaced by an elastic beam, where on its outer part different test components can be mounted for validation of aerofoil characteristics based on pressure measurements and testing of aerodynamic sensor and control systems. Besides the main beam, a counter weight



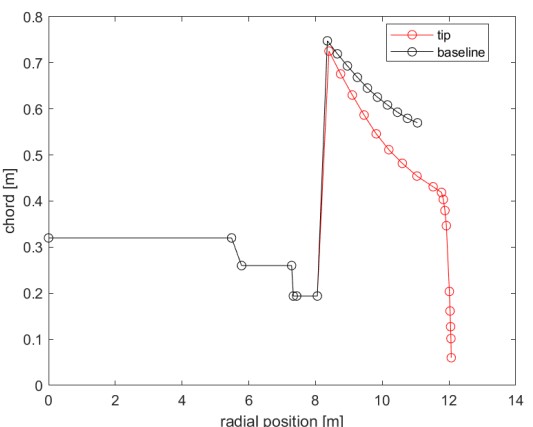
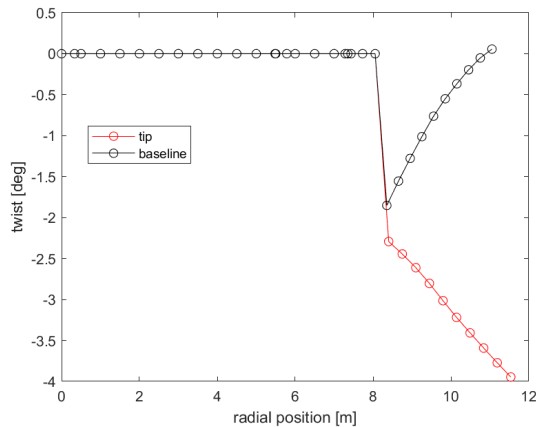

**Figure 4.** Planform of the optimized tip design compared to the reference.

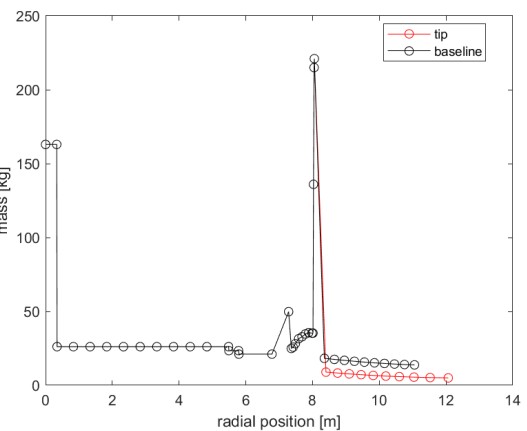
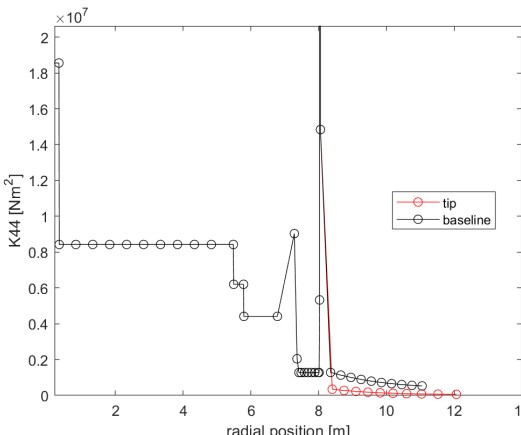

**Figure 5.** Mass and flapwise stiffness distributions of the tip compared to the reference.

is mounted to balance the beam and the aerofoil section. During the measurements the rotor shaft is motored and a frequency converter controls the rotational speed. The tip is mounted at the end of the boom via an adaptor (Fig. 7 and Fig. 8). The yaw angle, rotational speed and pitch angle are controlled to defined settings and logged, together with the wind speed and direction of the nearby met mast.

Four chord-wise bands of 1 mm inner diameter pressure taps are installed on the tip at spanwise locations of [9.09 9.79 10.49 11.18] m from the boom root. The 32 taps on each section are connected via tubing to one 1 psi and one 5 psi range DMT pressure scanners with an accuracy of 0.05 psi, located on the joint piece inboard of the tip root. Sets of strain gauges are installed at the sides of the spar cap and leading edge-trailing edge at two sections at spanwise locations of [9.8 10.6] m from the boom root. A 6-hole Pitot tube is also mounted at the joint piece inboard of the tip root, measuring the local inflow. The





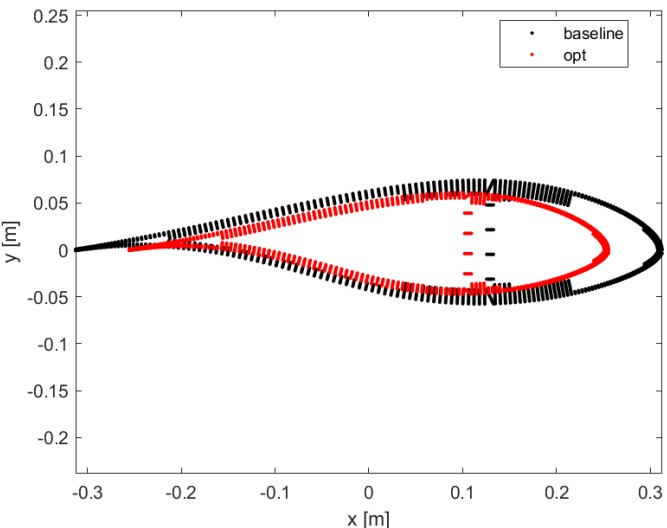

**Figure 6.** Internal structure of the mid-span section of the tip compared to the reference.

data acquisition (DAQ) system is based on cRIO from National Instruments. Finally, two cameras are mounted inboard of the
joint piece connected to the end of the boom.

The pressure distributions from the surface taps are numerically integrated into normal and chordwise aerodynamic forces on the local airfoil section reference frame, while an average pressure from the two nearby points is added to the trailing edge. The statistics for each measurement case are processed, together with the corresponding operation settings and inflow from the met mast.

The target result is the statistical distribution of the aerodynamic forces at each section for a set of statistic distributions of operation (wind speed, yaw, pitch). The parameters of the measured 300 s cases are shown in Table 2, with the averaged values based on the pitch angle settings in Table 3. Essentially, the average results of the 16 cases are condensed into the 4 idealized cases, which do not necessarily represent the physics but allow for a model-to-model comparison with low influence of the specific inflow condition.

**4 Aeroelastic simulations setup**

The time-domain aeroelastic simulations performed within the framework of the present study were orchestrated by the multi-body finite-element code HAWC2. All the computations shared the same structural modeling, which is described in Section 4.1. Several aerodynamic models were considered. In ascending order of fidelity, they are labelled in this document as:

- BEM: Corresponding to a BEM formulation, implemented as a built-in capability in HAWC2, and further described in
Section 4.2.



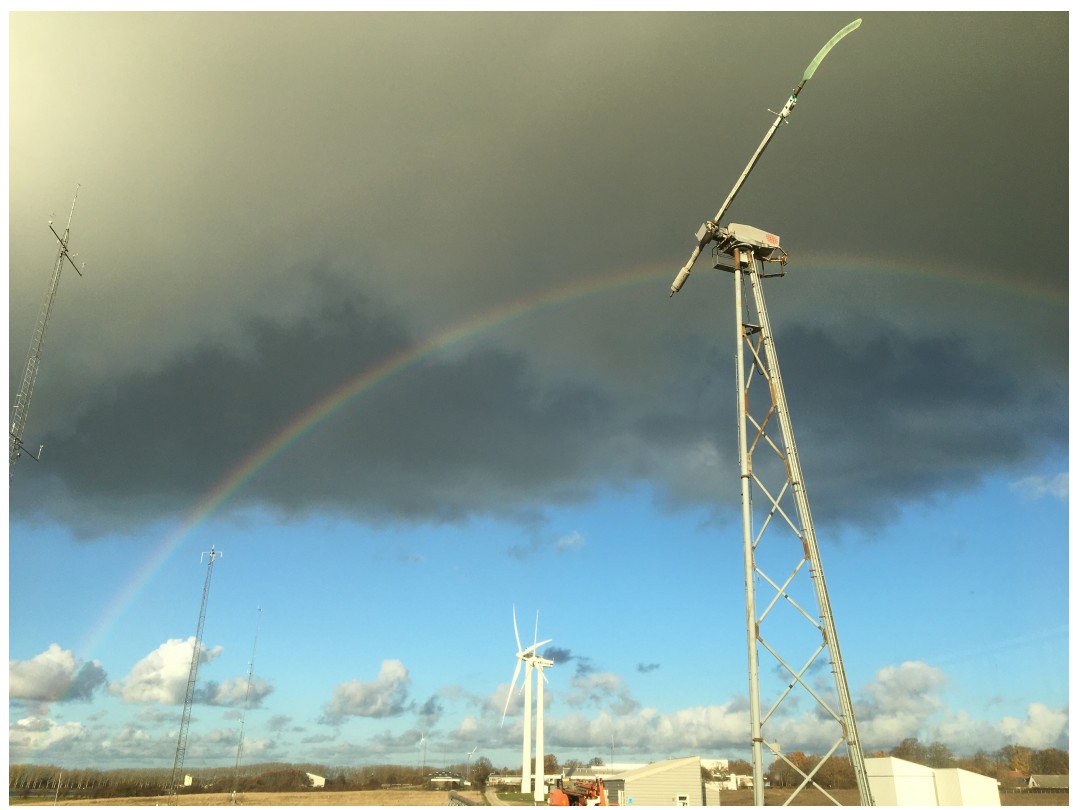

**Figure 7.** The tip mounted on the RTR and nearby met mast.

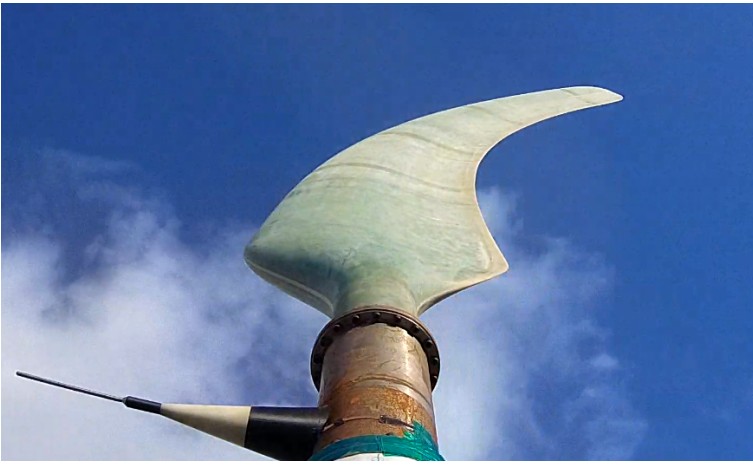

**Figure 8.** The tip as seen from the boom mounted camera.



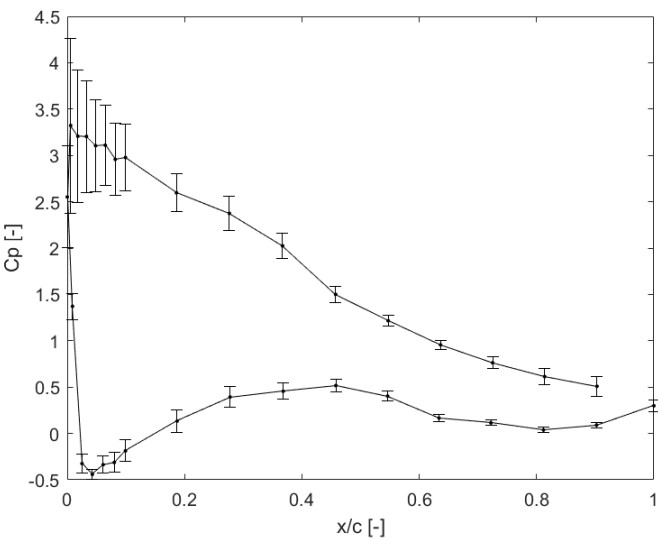

**Figure 9.** Example of measured average pressure coefficient at Section S2 of the tip for one case at 5° pitch. Error bars indicate standard deviation.

- NW: Corresponding to a vortex-based method coupling a near-wake model with a far-wake model, implemented as a built-in capability in HAWC2, and further described in Section 4.3.

- LL: Where the aerodynamic loading is computed with the stand-alone multi-fidelity vortex solver MIRAS, and integrated in the aeroelastic solution of HAWC2 through the loosely coupled scheme described in (Ramos et al., 2020). More details about the used aerodynamic model of MIRAS are included in Section 4.4.

- CFD: Where the aerodynamic loads are computed with the finite-volume Navier-Stokes code EllipSys3D, and integrated in the aeroelastic solution of HAWC2 through the loosely coupled scheme described in (Horcas et al., 2020). Additional details of the EllipSys3D model used for the present work can be found in Section 4.5.

### 4.1 Common structural model

All the presented methods were coupled with a common multi-body finite element HAWC2 model. For simplicity, the tower was considered to be stiff. Together with the ensemble of the boom and the tip, which was considered to be a single body, the shaft and the counter-weight were also modeled. The mechanical properties for the latter two components of the model are further described in Madsen and Petersen (1990). The boom and the tip were modeled by means of 32 bodies. Their mechanical properties, which were already summarized in Fig. 5, were introduced as a fully populated matrix.





**Table 2.** Parameters of all measured cases.

| nr | rotor speed [rpm] | pitch [deg] | wind speed mean [m s$^{-1}$] | T.I. [%] | yaw error mean [deg] |
|----|------------------|-------------|------------------------------|----------|----------------------|
| 1  | 26 | -5.0 | 4.7 | 16.3 | -1.4 |
| 2  | 26 | -5.0 | 4.5 | 12.2 | -1.5 |
| 3  | 26 | -5.0 | 4.8 | 18.3 | -2.7 |
| 4  | 26 | -5.0 | 4.4 | 20.6 | -2.5 |
| 5  | 26 | -0.1 | 3.4 | 17.3 | 2.5 |
| 6  | 26 | -0.1 | 5.0 | 16.1 | -1.1 |
| 7  | 26 | -0.1 | 4.9 | 13.8 | -3.5 |
| 8  | 26 | 0.0 | 4.0 | 16.2 | -9.3 |
| 9  | 26 | 5.0 | 5.0 | 15.7 | -2.1 |
| 10 | 26 | 5.0 | 4.6 | 15.6 | 2.1 |
| 11 | 26 | 5.1 | 4.2 | 14.4 | 4.7 |
| 12 | 26 | 5.1 | 3.3 | 17.2 | 10.4 |
| 13 | 26 | 10.0 | 4.9 | 17.8 | -4.2 |
| 14 | 26 | 10.1 | 4.8 | 14.3 | 4.4 |
| 15 | 26 | 10.1 | 4.2 | 12.7 | 0.9 |
| 16 | 26 | 10.1 | 3.7 | 10.5 | -8.2 |

**Table 3.** Parameters of measured cases averaged based on pitch angle.

| nr | rotor speed [rpm] | pitch [deg] | wind speed mean [m s$^{-1}$] | T.I. [%] | yaw error mean [deg] |
|----|------------------|-------------|------------------------------|----------|----------------------|
| 1  | 26 | -5 | 4.6 | 16.4 | -2.0 |
| 2  | 26 | 0  | 4.4 | 15.7 | -3.1 |
| 3  | 26 | 5  | 4.2 | 15.4 | 3.3 |
| 4  | 26 | 10 | 4.3 | 14.1 | -1.3 |

## 4.2 BEM aerodynamic model

The BEM method in the present study corresponds to the model described in (Madsen et al., 2020), that is implemented in the in-house finite element multibody aeroelastic code HAWC2 (Larsen and Hansen, 2007). The BEM method is implemented using a polar grid approach, which is capable of modeling turbulent inflow conditions. In addition, various submodels are included to model different effects, for example dynamic inflow, yawed inflow and unsteady 2-D airfoil aerodynamics. The blade is discretized radially into 80 sections using cosine spacing. Both the boom and the tip are included, while the cylindrical





boom is modelled with zero lift and drag coefficient of 0.8, as described in Sect. 2. The uniform inflow simulations have been performed for 50 s with a time step size of 0.01 s. The unsteady airfoil aerodynamic model (Hansen et al., 2004; Pirrung and Gaunaa, 2018) (usually referred to as the dynamic stall model) was used for all load cases. The Prandtl tip correction described in (Madsen et al., 2020) is included, which is not able to model the root vortex effect.

### 4.3  Near-wake aerodynamic model

The coupled near- and far-wake model (Madsen and Rasmussen, 2004; Pirrung et al., 2014, 2016, 2017a) is a computationally efficient vortex-based method. The near wake is defined as first quarter revolution of the trailed vortices from the own blade. It is modelled using non-expanding helical vortex filaments. The helix pitch is iterated within every time step using the indicial function approach and the steady-state induction is based on pre-calculated empirical functions that are fitted to the Biot-Savart law. The far wake is modelled using a far-wake BEM method that is without the tip-loss correction. The far-wake induction is calculated from a down-scaled thrust coefficient using a coupling factor. The near wake induction and the far wake induction are summed together to get the total induction. The coupling factor is calculated so that the rotor thrust is comparable to the one computed with the BEM method (Andersen et al., 2010; Pirrung et al., 2016). The near wake model was recently modified to model the blade sweep effects (Li et al., 2022), which also accounting for the curved bound vortex influence (Li et al., 2020). As for the BEM method, the unsteady airfoil aerodynamic model (Hansen et al., 2004; Pirrung and Gaunaa, 2018) is included for all load cases. As in the BEM method, the blade is also discretized radially into 80 sections using cosine spacing. And the cylindrical boom is modelled with zero lift and drag coefficient of 0.8, as described in Sect. 2. The uniform inflow simulations have been performed for 50 s with a time step size of 0.01 s. Unlike the BEM method that uses the Prandtl tip correction, NW models the near wake trailed vortices with helical vortex filaments and is able to model the root vortex effect.

### 4.4  Lifting-line aerodynamic model

Medium fidelity simulations have been carried out with the in-house vortex solver MIRAS, (Ramos et al., 2016, 2017). The lifting line (LL) aerodynamic model is used in the present study in combination with a hybrid filament-particle-mesh flow model (Ramos et al., 2019). In the LL model, the rotor blades are modelled as discrete filaments to account for the bound vortex strength and release shed and trailing vorticity into the flow. In the hybrid method, the vortex filaments are transformed into vortex particles which vorticity is later on interpolated into an auxiliary Cartesian mesh. The motion of the vortex elements is determined by a superposition of the free-stream velocity, the velocity induced by the blade bound vortex, the filament-wake and the particle-wake. The flow is governed by the vorticity equation, obtained by taking the curl of the Navier-Stokes equation. It describes the evolution of the vorticity of a fluid particle as it moves with the flow. Moreover, MIRAS has been recently modified to accurately account for blade curvature effects (Li et al., 2020). The dynamic stall model of Øye (1994) is used to account for the stall delay related to dynamic inflow changes seen by the airfoils. The coupling between MIRAS and HAWC2 (Ramos et al., 2020) accounts for the wind turbine structure dynamics, a collective pitch and torque control and the hydrodynamic loads. In a simplified manner, the coupling Python interface is in charge of transferring the blade aerodynamic loading computed by MIRAS, i.e. forces and moments at each aerodynamic section, to HAWC2. Moreover, the kinematics





of the blade computed by HAWC2, i.e. both the rigid body motion of the root and the blade axis translations and rotations at
every aerodynamic section, are transferred to MIRAS via the same framework. And in general, the coupling provides a common
framework between the different numerical codes, paving the way for a seamless comparison. A 12Rx4Rx4R Cartesian mesh
as been employed in all cases, with a constant spacing of 0.5 m in the x, y and z directions which adds up to a total of more
than 2 million cells. The bound vortex is discretized with 80 straight segments with a constant spacing. Both the boom and the
tip are included in the LL model. All uniform inflow simulations have been performed for 200 s with a time step size of 0.008
s, adding to a total of 25000 time steps. A free boundary condition is used in all directions. Moreover, an eight-order stencil
is used to spatially discretize the vorticity equation, a particle re-meshing is forced every time step to maintain a smooth field,
and a periodic re-projection of the vorticity field is applied to maintain divergence free field.

## 4.5    EllipSyS3D aerodynamic model

Higher fidelity simulations were performed with the three-dimensional computational fluid dynamics software EllipSys3D
(Michelsen, 1992, 1994; Sørensen, 1995). EllipSys3D is a multiblock finite-volume code for structured grids, accounting for a
wide range of modeling capabilities. In the present work, the incompressible Reynolds-Averaged Navier-Stokes (RANS) equa-
tions were solved in general curvilinear coordinates. The solution was advanced in time with a dual time stepping method, using
a time step of 4 ms. To accelerate the convergence of the solution, grid sequencing was used. The k-$\omega$ SST turbulence model
(Menter, 1994) was selected, due to its performance when dealing with airfoil flows. Two distinct sets of simulations were
carried out. One assumed fully turbulent flow, while the other accounted for a correlation-based transition model (Sørensen,
2009). These two sets of computations are labelled in the present document as `turb` and `trans`, respectively.

The deflections of the boom centerline and the mounted tip centerline, computed by HAWC2, were introduced in the CFD
solution during run time. These deflections were subsequently applied to the surface grid through a spline interpolation. The
surface grid deflection was then smoothed into the inner domain through a mesh deformation algorithm, based on the distance
to the blade surface. Rotation was simulated by applying a rotational motion to the full computational grid, as a solid body. At
every time step, the CFD loading was computed and injected into the HAWC2 solution. This is done by integrating the pressure
and friction loads (including forces and moments), in a series of sectional planes which are normal to the local blade axes. The
location of such sectional cuts was forced to correspond to the position of the aerodynamic sections that were defined in the rest
of the methods included in this work. For more details about the EllipSys3D and HAWC2 aeroelastic coupling implementation
, the reader is referred to (Horcas et al., 2020).

The grid was generated in two consecutive steps. First, a structured mesh around the cylindrical boom and the tip surfaces
was generated with the Parametric Geometry Library (PGL) tool (Zahle, 2019). A total of 128 cells were used in the spanwise
direction, and the chordwise direction was discretized with 256 cells (with 8 of them lying on the trailing edge). Secondly, the
surface mesh was radially extruded with the hyperbolic mesh generator Hypgrid (Sørensen, 1998) to create a spherical volume
grid. A total of 128 cells were used in this process, and the resulting outer domain was located approximately 30 m away from
the tip surface. A boundary layer clustering was taken into account, with an imposed first cell height of $1 \times 10^{-5}$ m, in order
to target y$^+$ values lower than the unity. The resulting volume mesh had a total of 5.2 million cells. An inlet/outlet strategy





was followed for the boundary conditions of the outer limit of the domain, and both the boom at the tip itself were modeled as walls. A sketch of the ensemble of the boundary conditions is depicted in Fig. 10, together with a visualization of the mesh.

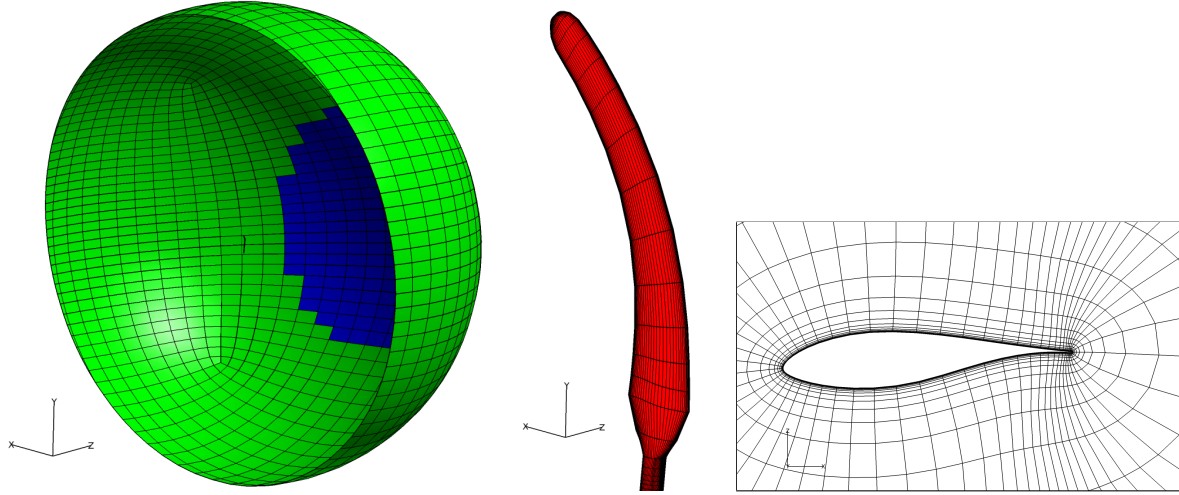

**Figure 10.** Visualization of the EllipSys3D mesh. For clarity, only one out of every four grid lines is shown. *Left*: overview of the boundary conditions distribution. Half of the spherical domain was not depicted, and the freestream velocity vector was aligned with z axis. Green colour corresponds to inlet, and blue to outlet. *Middle*: detail of surface mesh around the walls, here depicted in red. *Right*: cross-sectional mesh around the tip at half of the projected length.

## 5 Comparison of test and simulation results

This section contains the comparison of measured loads with the aeroelastic loads predicted using the varying fidelity aerodynamic models. Because no wake rake was mounted on the rotating rig only the measured forces normal to the chord are available. Insufficient wind speed measurements were available to accurately estimate shear coefficients, so no mean shear profile is present in the simulations. The effect of shear is however assumed to be minor due to the small rotor diameter of the rotating test rig. Most of the simulations shown here use transitional polars. It has been investigated that the comparison of fully turbulent CFD and LL using fully turbulent polars is consistent and this is briefly demonstrated. But generally the loading was found to be underpredicted using fully turbulent polars and fully turbulent CFD. These results are omitted here for brevity.

The section is organized as follows: In Section 5.1, aeroelastic simulations of all fidelity levels are compared to the mean experimental results. A comparison of sectional loads as function of pitch angle is included and comparisons to the earlier wind tunnel tests of a scaled geometry in Barlas et al. (2021b) are made. In Section 5.2, the effect of turbulence on the spanwise load distribution and the distributed standard deviation of the loading is evaluated using the BEM, NW and LL solvers.





### 5.1 Uniform inflow

Unless otherwise stated, the results shown here are averaged from four simulations at the four slightly different operating conditions for each pitch angle, see Table 2.

**Spanwise load distribution**

The averaged normal force from measurements and simulations and the averaged simulated chordwise force are shown in Fig. 11. Similar observations can be made for -5° and 0° pitch angle: Aside from an outlier at the most outboard section for the -5° pitch case and a generally slight underprediction, the codes capture the measured normal forces reasonably well. The shape of the normal force is predicted similarly well by NW, LL and CFD, which all predict a smaller slope than the BEM simulations. The largest difference in slope is found to be inboard, where there is a clear effect of the root vortex visible in the results of all codes except BEM. Towards the tip, the normal loading predicted by LL and NW is larger than the loading predicted by CFD, which is largely due to the smoothed chord in the CFD simulations, see Fig. 4. The CFD simulations of the chordwise force show a large peak towards the tip, which will be investigated later in this section.

At 5° pitch, no peak is observed in the chordwise loading of the CFD simulations. The measured normal force agrees very well with the predictions by LL and NW. The beginning of stall is seen to lead to a less smooth load distribution of the NW simulations outboard of 10.5 m span. This is less visible in LL, possibly due to the use of a vortex core and a less fine point distribution towards the tip.

At 10° pitch, shown in the last row of Fig. 11, some stall delay becomes visible in the CFD simulations and the experiment, leading to much higher normal forces than predicted by the codes that rely on airfoil data. The NW and, to a lesser extent, LL simulations now show non-smooth force distributions along the whole span due to stalling flow. Because the operation here is close to maximum lift, at a small $dC_L/d\alpha$ slope in the polars, the influence of the root vortex and tip vortex becomes much smaller in the codes using airfoil data. Therefore the difference between BEM and LL/NW is smaller than for the lower pitch angle cases.

**Sectional loads as function of pitch angle**

The purpose of Fig. 12 is to evaluate the predicted and measured slopes of the normal forces as function of pitch angle. This enables the comparison of trends with the wind tunnel work in Barlas et al. (2021b). At Section S1, the effect of missing root vortex in the BEM simulations is clearly visible, causing an overprediction of the loading in the linear region. This is in good agreement with the data presented in Barlas et al. (2021b). The slope in the experiment and CFD is linear up to 10°, while the codes using airfoil data predict the onset of stall.

At Sections S2 and S3, the experiment shows signs of stall towards 10°. The CFD-based simulation also predicts a linear behavior, while the other codes stall too early. At Sections S3 and S4, all airfoil data based codes predict almost the same loading, while the experiment shows a lower slope than all codes in the attached flow region, especially at S4, and a clear stall at 10°.







**Figure 11.** Comparison of normal (left) and chordwise (right) load distribution obtained from simulations and measurements. Results shown are averaged from four simulations per pitch angle, see Table 2.

The generally very good agreement between NW and LL computations in attached flow was also observed in the previous
comparison with wind tunnel measurements. At Section S4, the agreement is improved in the present work because the swept



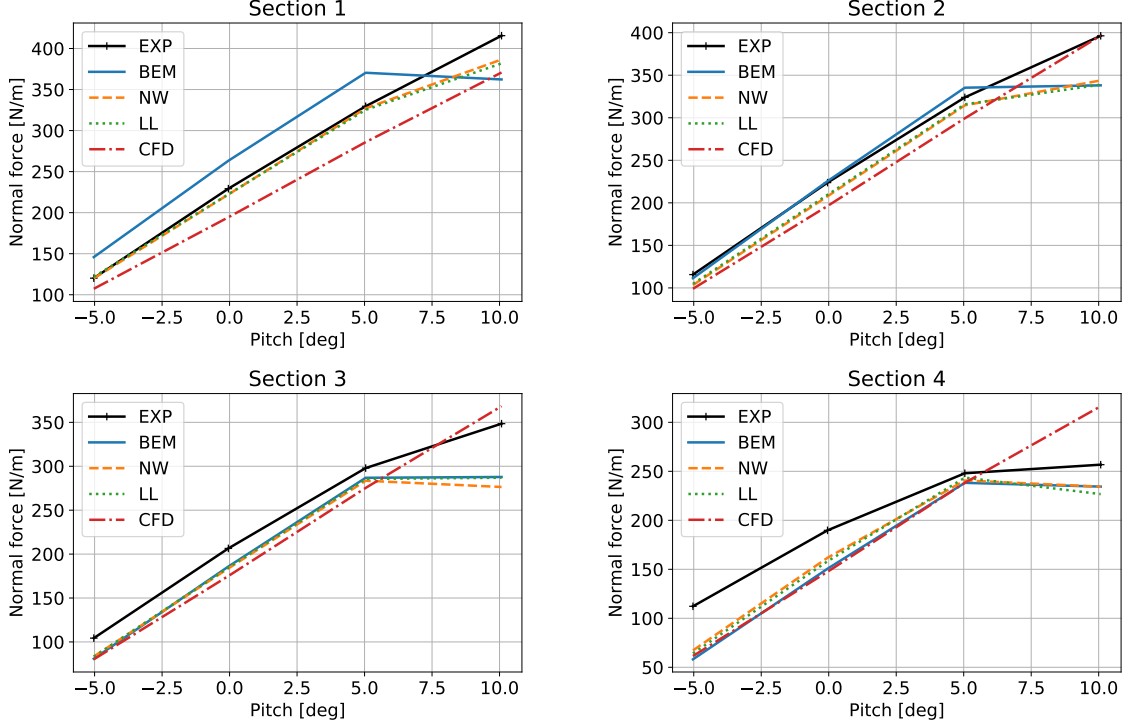

**Figure 12.** Comparison of normal force from simulations and experiment as function of pitch angle. Results shown are averaged from four simulations per pitch angle, see Table 2.

tip shape is taken into account by the NW model due to the modifications described briefly in Sect. 4.3. In the present campaign, the measured loads are generally higher than the predicted loads. In the wind tunnel campaign, the agreement between measurements and simulations was better, likely due to the more accurate knowledge of the inflow conditions. The better performance in stall predicted by CFD compared to the measurements and the inability of the airfoil data based models to predict

performance in stall are also consistent with the wind tunnel campaign in Barlas et al. (2021b).

**Deflections**

The torsional and flapwise deflections for Case 3 and 4 are shown in Figure 13. Because all aerodynamic models are coupled to the same structural sover, the very similar aerodynamic forces in Case 3, see Figure 11 cause very similar structural deflections. In Case 4, the delayed stall predicted by the CFD leads to comparably larger deflections. The deflections are generally small,

but the torsional deflection will have an influence on the mean loading due to its close relationship with the angle of attack. The agreement of the predicted deflections in Case 1 and 2, the plots of which are not included here for brevity, is very similar to Case 3 but at a lower overall level due to the smaller aerodynamic forcing.





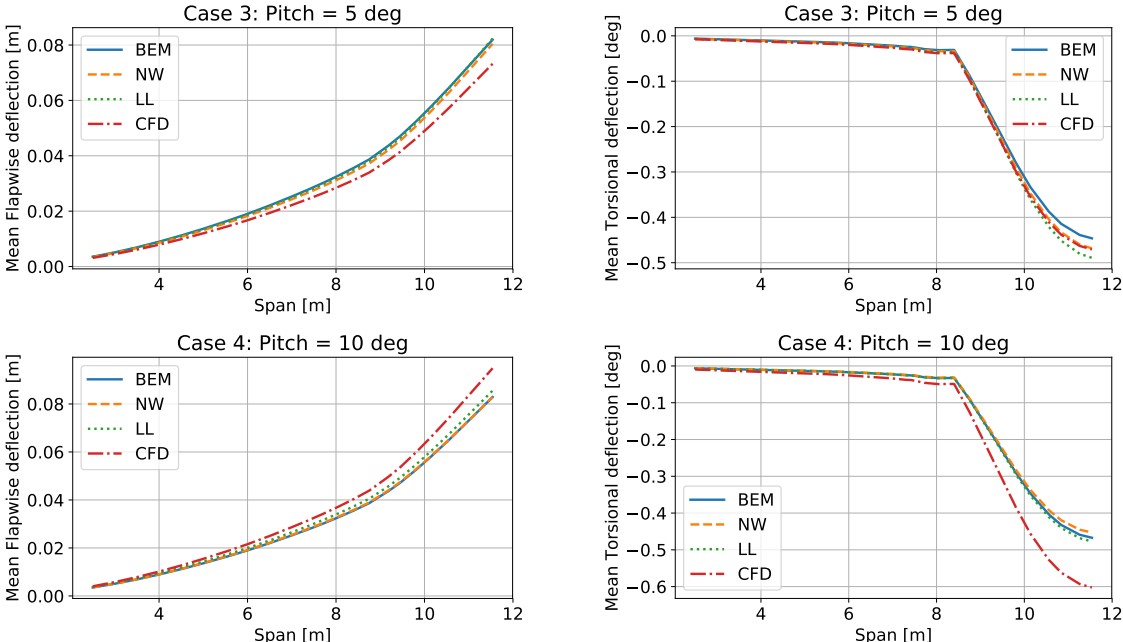

**Figure 13.** Comparison of mean deflections for Case 3 and Case 4. The torsional deflection is given about the pitch axis.

## Investigation of 3D flow at the very tip and transitional versus turbulent simulations

Figure 14 sheds some light on the origin of the peaks in the chordwise force distribution predicted by CFD in Fig. 11, and
compares transitional and turbulent simulations. The LL results are shown here, because they represent the highest fidelity
code using airfoil data in this study. The left column shows the resultant force due to the combination of the normal force and
chordwise force, and the right column shows the angle of the resultant force with respect to the airfoil chord. It becomes clear
that the rotation of the resultant force in the CFD results towards the tip causes the peak in the chordwise loading. This rotation
is probably due to the highly three-dimensional flow at the tip, as it is illustrated in Fig. 15 for the particular case of Case 3.

LL is unable to predict the near-tip direction change of the load, and actually these angles and forces would not be possible
to achieve based on airfoil data, because there is a significant spanwise flow in the CFD simulations. Along these lines, it could
be speculated that one of the factors explaining why all the other methods showed higher loading when compared to CFD
could also rely in this three-dimensional behaviour.

As already mentioned, there seems to be generally larger tip loss in the CFD simulations than in the LL simulations. This is
in part due to the rounded tip geometry (see Fig. 4), and was a common feature for both turbulent and transition simulations.
In attached flow (see Case 3 in Fig. 14), the differences between laminar and turbulent profile data agree very well with the
differences in the CFD simulations between transitional and fully turbulent flow. The agreement becomes worse at a pitch
angle of 5°, where light stall is already present. At 10° the flow is stalled in LL, which leads to very small differences between



eawe
european academy of wind energy

WIND
ENERGY
SCIENCE
DISCUSSIONS

**Figure 14.** Comparison of resultant force magnitude (left) and angle (right) between LL and CFD for cases 3, 9 and 13 from Table 2. Clearly the spikes in chordwise force towards the tip in the CFD results shown in Fig. 11 are due to a rotation of the force.

the transitional and turbulent simulations. In the CFD simulations, the stall is delayed and thus the difference between laminar and turbulent flow is much larger.

### 5.2 Turbulent inflow

Simulations accounting for an inflow turbulence that matches the measured one during the experimental campaign have been carried out. The averaged cases based on pitch angle defined in Table 7 have been numerically reproduced with three different



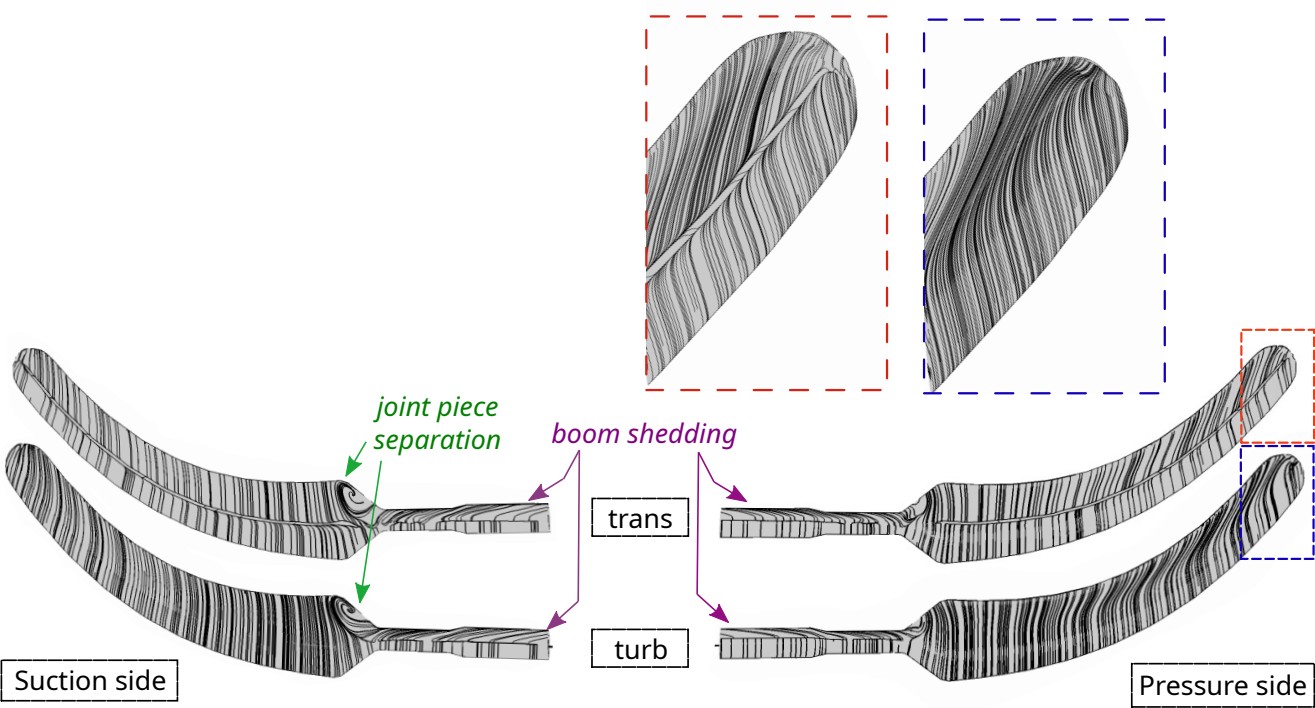

**Figure 15.** Surface-restricted stream lines computed with the CFD method for Case 3, corresponding to a pitch of -5°. Both `trans` and `turb` results are presented, based on the solution of the last computed time step.

fidelity simulations, i.e. BEM, NW and LL. Fully resolved CFD simulations have not been carried out due to the significantly
high computational requirements of such cases, beyond the resources allocated for the present work.

First of all, the turbulence generator of Mann (1998) has been used to create four turbulence boxes with an objective turbu-
lence intensity of 15.4 % which matches the average turbulence intensity of the four cases. Seeds number 202, 302, 402 and
502 have been used to account for different turbulence realizations. The generated boxes have a size of 4096×32×32 cells in
the stream-wise, vertical and lateral directions respectively with a constant cell size of 2 m. The following constants have been
used to account for land based turbulence generation $\alpha\epsilon$ = 1.0, L= 40 and Gamma = 3.9.

Cases number 1 to 3 in Table 7 have been modelled in LL, each one including a different pitch angle, wind speed and yaw
angle. Case 4 was not ran with turbulence due to the high angle of attack (AoA), which causes stall and leads to issues with the
NW and LL vortex solvers. All four generated turbulent boxes (4 seeds) are run in each one of the LL setups, adding up to 12
simulation cases. LL simulations with and without the rotating test rig are performed, adding up to a total of 24 cases.

In MIRAS, the turbulent boxes are transformed into a particle cloud by computing the curl of the velocity field. The turbulent
particles are released one diameter upstream the rotor plane and interact freely with the turbine wake if existent. The vortex
solver accounts for turbulence development as it convects downstream towards the rotor plane. In the simulations without
a turbine, the local velocities are calculated every time step at the rotor plane position, a 64×64 mesh with a cell size of





approximately 1.5 m is used for the velocity extraction. Such velocity field is loaded in the NW and BEM simulations, allowing
to account for the same turbulent structures simulated in MIRAS. In this way it is possible to closely mimic the turbulence seen
by the turbine in MIRAS, although the influence of the turbine and its wake on the inflow turbulence field can not be accounted
for in the lower fidelity models.

All codes (BEM, NW and LL) simulate each seed for 900 s at a time step of 0.01 s. The initial 100 s are discarded in the
postprocessing.

In the following, the mean and standard deviation of the loads from the turbulent simulations are compared to the experi-
mental values.

**Spanwise mean loading distribution**

The spanwise mean loading in the normal and chordwise direction obtained from measurements and simulations is shown in
Fig. 16.

The shaded area in Fig. 16 represents the standard deviation of the mean values of the results obtained using four different
turbulence seeds. They have an almost constant width along the span. This shows that the different seeds lead mainly to different
offsets of the mean load distribution and not to different slopes. An exception is the region towards the tip, where the shaded
areas narrow, especially for the chordwise force. The error bars in the experiments represent the standard deviation of the mean
values obtained from the experiment. The averaged experimental values differ both in operating points (see Table 2) and in
turbulent realization. Therefore the standard deviations of the measured mean load distributions is not directly comparable to
the standard deviations for the simulations, which is only due to different turbulent seeds. The observations regarding slopes
of the loading and comparison to experiment are similar to the uniform inflow cases shown in Fig. 11. The NW and LL results
show an excellent agreement except at the very tip, where the NW method predicts higher loading. The BEM results have a
steeper slope with higher loading inboard due to the missing root vortex effect and lower loading outboard due to the missing
effect of the backward sweep on the tip loss.

Generally, the spread between the results for different turbulence seeds indicates that a large part of the deviations between
experiments and simulations may be explained by variations between turbulence realizations, with an exception of the outmost
section at a pitch of -5°.

**Standard deviation of spanwise loading**

The distribution of the standard deviations of the normal loading and chordwise loading are shown in Fig. 17. The shaded
area represents the standard deviation of the four standard deviations of the simulations using four turbulent seeds. As above,
the error bars in the experiments represent the standard deviation of the mean standard deviations values obtained from the
experiment, where the four measured time series differ in both mean operating point and turbulence. This is not directly com-
parable to the simulations, which were performed at the mean operating conditions to reduce computational cost. Therefore,
the error bars from the experiment, which include variations due to mean operating point variation and turbulence realization,
are generally wider than the shaded areas from the simulations, which vary only due to turbulence.



**Figure 16.** Comparison of normal and chordwise force from **turbulent** simulations and experiment. Results shown are averaged from four turbulent seeds per pitch angle, using the mean operating conditions as shown in Table 3 up to a pitch angle of 5°. The shaded areas show the standard deviation between the mean values of the four simulations per pitch angle to indicate the variability due to turbulent seed.

The measured standard deviation of the normal force is generally similar to the simulated standard deviation. An exception is Case 2 at roughly zero degrees pitch, where the simulated standard deviations are about 20% larger than the measured values. The slope of the standard deviation of the loads as function of the span seems to be overpredicted by BEM compared to experiment. The NW and LL results are in better agreement with the measured data. This is likely due to the dynamic modeling



**Figure 17.** Comparison of standard deviations of normal force from **turbulent** simulations and experiment. Results shown are averaged from four turbulent seeds per pitch angle, using the mean operating conditions as shown in Table 3 up to a pitch angle of 5°. The shaded areas show the standard deviation between the standard deviations of the four simulations per pitch angle to indicate the variability due to turbulent seed.

of the root vortex influence in the NW and LL simulations, which clearly reduces the standard deviation of the loading at the inboard part. The root vortex influence is generally more prominent in the chordwise force, because the induced vorticity causes a change in AoA that changes both the magnitude and direction of the aerodynamic forces. The magnitude affects both





the normal and chordwise forces, while a change of the angle mainly leads to differences in the chordwise force. The effect of
beginning separation is clearly visible in the NW simulations at 5°, especially close to the tip.

The shaded area, which represents the spread in standard deviation between turbulence seeds, agrees very well between LL and NW, with the BEM predicting a much larger spread in Case 3 (5° pitch). It is unclear why the four different seeds lead to almost exactly the same standard deviations of the chordwise force at a pitch angle of zero degrees, even though the standard deviation of the normal force varies with turbulence seed. But the effect is consistent across the model fidelities, and it was
confirmed that the four seeds lead to four different time series of the chordwise loading, which happen to have almost exactly identical standard deviation.

## 6   Conclusions

The aeroelastic response of a swept tip is investigated for application to wind turbine tip extensions by controlled field testing in the outdoor rotating rig at the Technical University of Denmark (DTU). The swept tip shape in focus is the result of design
optimization, focusing on locally maximizing power performance within load constraints compared to an optimal straight tip. The tip model is instrumented with spanwise bands of pressure sensors and is tested in atmospheric inflow conditions. A range of fidelity of aerodynamic models is utilized to simulate the test cases and are compared with the measurement data, namely a blade element momentum (BEM) model, a coupled near- and far-wake model (NW), a lifting-line hybrid wake model (LL), and a fully resolved Navier-Stokes computational fluid dynamics (CFD) simulations. The first simulations tackled a series of
idealized inflow conditions, that were obtained by averaging several time windows of the experimental data. Results show that the measured mean normal loading can be captured well with the vortex based codes and the CFD solver. The CFD solver seemed to generally underpredict the measured mean loading for these idealized conditions in attached flow. However, this higher fidelity method computed a similar stall delay as seen in the measurements at high angle of attack. Similar trends as in earlier wind tunnel measurements were observed when plotting the measured and simulated loading against the pitch angle.
The CFD solution shows a highly three-dimensional flow at the very tip that leads to large changes of the angle of the resultant force with respect to the chord at the very outboard part of the curved tip. These angle changes are not able to be predicted by any model using 2-D airfoil data. No measurements were available at such outboard stations and therefore we were not able to validate this phenomena with measurements. In a second stage of the analysis, the influence of turbulence on the definition of the ideal cases was addressed. Simulations with four different turbulence realizations indicated that a large part of the
deviations between measured and simulated mean loading by the higher fidelity codes can be due to seed to seed variations. These turbulent simulations show that the measured standard deviations of the normal force match those predicted by the vortex codes well. There are some deviations when comparing to the BEM simulations, especially towards the root section. Future work should focus on full scale validation of aeroelastically optimized tip shapes, with focus on further enabling structural tailoring features and topologies, and possible combination with active aerodynamic add-on features.





*Code and data availability.* Pre/post-processing scripts and data sets available upon request. The codes HAWC2, MIRAS and EllipSys3D are available with a license.

*Author contributions.* Thanasis Barlas performed the tip design optimization, contributed to the model preparation, performed the tests and contributed to the model setup and comparison. Georg Pirrung contributed to the tip design optimization and model setup and comparison. Néstor Ramos García contributed to the model setup and comparison. Sergio González Horcas contributed to the model setup and comparison. 350 Ang Li developed the implemented the updated Near Wake model utilized in this work and contributed to the comparison. Helge Aagaard Madsen contributed to the tip design optimization and the model preparation and testing.

*Competing interests.* No competing interests are present.

*Acknowledgements.* This research was supported by the project Smart Tip (Innovation Fund Denmark 7046-00023B), in which DTU Wind Energy and Siemens Gamesa Renewable Energy explore optimized tip designs. The following persons have also contributed to the presented 355 work: Flemming Rasmussen, Niels N. Sørensen, Frederik Zahle, Peder B. Enevoldsen and Jesper M. Laursen.



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
