# Peer review of "Atmospheric rotating rig testing of a swept blade tip and comparison with multi-fidelity aeroelastic simulations"

_Wind Energy Science, 2022_

## Referee Comment (RC2)

[referee-annotated manuscript omitted]

---

## Author Comment (AC1)

**Authors' response to review of article wes-2022-35-RC1/RC2**

RC1

**RC1_1:**

The authors wrote a follow-up paper regarding blade tip design. The first parts, cited in reference in the present work, deal with tip design optimisation and wind tunnel test. The results presented in the current article, show a comparison between: numerical simulations of several level of fidelity and field test using the Rotating Test Rig available on the DTU campus. Ideal and turbulent simulations were performed for the all numerical set-ups except the CFD due to high resources required. Comparison by mean of spanwise loading is made. An interested behaviour noted, is the highly 3D flow at the tip captured changing the AoA and force direction in ways that BEM, LL and NW cannot capture yet.

**AC1_1:**

The authors would like to thank the reviewer for their time and greatly appreciate their feedback and suggestions to improve the article.

**RC1_2:**

The tip model design chapter (chapter 2), is unnecessarily long since it has been referenced several times in the authors previous work. Shortening this chapter will allow for more detailed results (see comment 10 and 12). Similarly, the Figure 2 to Figure 5 are interesting but similar plots are already cited in previous works.

**AC1_2:**

The design optimization methodology is slightly different in this work, since it included the structural layup design variables and different constraints and objective evaluation due to the RTR setup compared to a full scale case, but similar to the wind tunnel model design. The figures need to be there since the setup is fundamentally different from the full scale blade and wind tunnel model setups presented before. So, the section has not been edited. Also note that another review comment (AC_2_2) requests the expansion of this section, justifying the interest on the contents covered.

**RC1_3:**

The Figure 1 should be kept for the sake of showing the object of study, but the rules regarding technical drawing are not followed. The dimensions should be above (or below) arrowed lines. The way it currently is, is confusing.

**AC1_3:**

The figure has been updated, with the dimensions removed, since the information appears in other figures.

**RC1_4:**

In the chapter 4.5, the Figure 10 has already been used in previous work. Instead, a plot detailing the sectional cuts and axis convention used as detailed in lines 167 to 169 could have a better value.

**AC1_4:**

Note that Figure 10 (original manuscript numbering) has not been published elsewhere. While the generated mesh resembles the one included in Barlas et al. 2021, the present configuration is slightly different (a spherical domain is meshed, and a body-fitted grid around the boom is considered). Therefore, we believe it is important to keep that figure as originally designed. Following the suggestion of the reviewer, we have included a new figure illustrating the position of the aerodynamic sections. That corresponds to Figure 10, based on the numbering of the revised manuscript.

**RC1_5:**

Line 185, "Most of the simulations shown here use transitional polars" does it mean that it was calculated using 2D or 3D CFD using a transition model ? The numerical simulations were not all done for the same amount of time, was a sufficient level of convergence achieved for all solvers ? Why not, using a similar amount of time for all?

**AC1_5:**

All tools except for CFD used wind tunnel polars with free boundary layer transition (clean) as described in the text.

For the particular case of the uniform inflow the involved numerical methods were run with a different total simulated time. To check the impact of that parameter on the presented results, preliminary simulations were conducted. This was done in an independent manner, so that each solver user ended up defining a method-related total simulated time. As suggested by the reviewer, we could have taken the maximum of the considered values, in order to use a common total simulated time for all the simulations. However, we did not consider that step to be strictly necessary, as what is crucial is that all the simulations were run "long enough" (so that there are no effects from the initial transient or the flow development).

**RC1_6:**

It is mentioned, that the Prandtl tip correction is used. How confident are you about the correction applied to a highly swept blade tip ? Has the correction model been updated to account for the tip shape ?

**AC1_6:**

The tip correction is only applied in BEM and it is not expected to perform well in highly swept geometries. Text has been added to explain this.

**RC1_7:**

The 3D CFD simulations show that the root vortex is only located at the transition part between boom and tip. Was it seen in the experiment, using the installed cameras and tufts for instance ? Was there any radial flow noted ?

**AC1_7:**

Unfortunately, there was no flow visualization performed during the tests, so we only rely on the CFD results to explain the 3D flow at the boom transition and tip. Some text has been added to mention this.

**RC1_8:**

Chapter 5.1 "Spanwise load distribution". For the -5° case, I would not say that the forces are "well captured". The numerical model show a clear decrease along the span, while the experiments show the same load level for the first 2 points and a small decrease for the third one. I don't think the trend is captured. Moreover, with only 3 experimental points (the tip being an outlier) it is difficult to conclude in that specific case. However, the other cases show great similarity between the simulations and experiments.

**AC1_8:**

The text has been updated to reflect this observation.

**RC1_9:**

For clarity, maybe use a single legend for all plots for the Figure 11. The same can be said for all the figures comparing results (Figure 12, 13, 14, 16, 17). The authors mentioned several times wind tunnel tests, it will be interesting to include the wind tunnel results in the plots, to see the whole picture: simulation, wind tunnel, small scale field tests. Similarly, the pressure coefficient plot (Figure 9) could be compared with 3D CFD results, especially for the Section 4 (if available).

**AC1_9:**

As the legends of Figure 11 do not interfere with the curves, we prefer to reproduce them in every plot (consistently to the rest of the figures of the manuscript).

In our opinion it would be confusing to directly compare the results with the wind tunnel test, since there is no rotation and flow velocity change along the span.

The pressure distribution could not be compared to the CFD results for a technical reason. Indeed, at the moment of conducting the simulations, the pressure distribution output was not available for the CFD-based aeroelastic simulations. While implementing such a feature was relatively straightforward, the available resources for the present manuscript did not allow for performing the required development.

**RC1_10:**

The Figure 12 seems to show that the CFD is "less accurate" than the LL and NW. For the section 1 it is as far from the experiments as the BEM in opposite direction. The other sections are more in line with the rest. Has it been investigated ? Can the swept shape be not suitable to the meshing strategy adopted here ?

**AC1_10:**

We believe that it is not fair to conclude that CFD is less accurate compared to LL/NW, since their difference is an offset, which could vary with all related uncertainty in the measurements. Along these lines, it should be noted that the inflow turbulence had a remarkable influence on the computed loads, and that effect was omitted in the CFD-based runs. In other words, the idealization of the uniform inflow conditions implied a relatively high degree of simplification, so that a one-to-one comparison with the experimental data could be misleading.

Based on the experience of the group running the EllipSys3D solver, there is nothing that could indicate that the mesh topology is not suited for the performed simulations. Indeed, the elements of both the

surface grid and the volume mesh had an acceptable quality, despite the curvature of the blade. Note that the local angles introduced by this curvature on the near-wall mesh remain relatively low, so that the hyperbolic mesh generator was able to produce a high quality grid even in that region.

**RC1_11:**

The Figure 17 and associated description doesn't not add much information since the standard deviation is already present on Figure 16. This paragraph could be replaced by the comparison with wind tunnel data and/or presentation of some aerodynamic parameters along the tip depending on the simulation method and field data (if available): induction, angle of attack, power coefficient,…

**AC1_11:**

The standard deviation presented in Figure 16 and 17 (Figure numbers corresponding to the original submission) are different. In Figure 16, central line is the mean of the normal forces (mean_seed(mean_time($F$^s(t)))), where s indicates the turbulence seed or the case number. The shaded areas indicate how much the mean values of the forces vary with different turbulence seeds (std_seed(mean_time($F$^s(t)))). It is included to give an indication of how much offsets in the mean values may be caused by different turbulence realisations. The main conclusion from Figure 16 is that the seed-to-seed variations between turbulent simulations are mainly causing offsets in the mean loading, not as much changes in slope.

Figure 17, on the other hand, compares the means of the std (mean_seed(std_time($F$^s(t)))) and standard deviations of the standard deviations of the forces (std_seed(std_time($F$^s(t)))) for the different seeds and measurement cases. So Figure 17 compares the measured and simulated dynamic behavior. It shows that the root vortex clearly causes a reduction in the standard deviation of the normal force at the most inboard section both in the measurements and in the higher fidelity simulations. This effect is missed by the BEM simulations and disappears in the 5 degree case which is closer to stall.

We have added some definitions to this section to clarify what is shown in the plots.

RC2

**RC2_1:**

The paper presents a thorough comparison of field, load measurements of a swept blade tip, attached to a rotating test rig, against predictions by varying fidelity aeroelastic models. All simulations are using the same structural module but aerodynamic modules of different fidelity covering the whole span of the existing state-of-the-art aerodynamic models (BEM, hybrid BEM and vortex near wake, lifting line vortex wake, CFD). The work presented in the paper is novel, interesting to the research community, well documented and in the reviewer opinion deserves publication in WES journal.

**AC2_1:**

The authors would like to thank the reviewer for their time and greatly appreciate their feedback and suggestions to improve the article.

**RC2_2:**

My only major comment concerns section 2. The information presented in this section is very condensed: The brevity of the presentation does not allow design optimization objectives to become clear

**AC2_2:**

The design optimization methodology has essentially been presented in earlier work, although slightly different here, since it included the structural layup design variables and different constraints and objective evaluation due to the RTR setup compared to a full scale case, but similar to the wind tunnel model design. So the section has not been edited. Note that another review comment (AC_1_2) requests the shortening it instead of expanding it.

**RC2_3:**

Some of the figures presented are not well explained in the text (for example figure 2)

**AC2_3:**

Text has been added in the caption of Figure 2, in order to explain it.

**RC2_4:**

In many occasions modeling aspects (for example drag coefficient used for cylinders) are mixed up with design aspects (type of airfoils used and Re at which polars are produced).

**AC2_4:**

Unfortunately this can't be further improved due to the condensing of the design section.

**RC2_5:**

The reported increase in power achieved (19.58%) is too high and needs some further elaboration.

**AC2_5:**

Since the RTR is a powered setup the local power changes can not be translated to any meaningful full scale turbine application, but here are simply used in the design optimization. Text has been added to clarify this.

**RC2_6:**

[line 45] This is a modelling aspect. The design aspect is that you have used constant airfoil shape.

**AC2_6:**

We agree with the observation of mixing design and modelling aspects, but the polar details need to be mentioned here since this affects the design of the tip. Since the section is condensed, we can not avoid this overlapping.

**RC2_7:**

[line 47] Separate design from modeling aspects.

**AC2_7:**

See previous answer.

**RC2_8:**

[line 49] for the sake of completeness please include tip speed ratio value.

**AC2_8:**

The text has been updated.

**RC2_9:**

[line 49] What is the objective of the aeroelastic optimization.

**AC2_9:**

The objective is to locally maximize power performance within load constraints compared to an optimal straight tip. The text has been updated.

**RC2_10:**

[line 50] This is unrealistically high. Unless of course the tip speed ratio at which you optimize is very far from the optimum of the baseline blade. If this is the case then what is the point of doing that?

**AC2_10:**

As replied above, the RTR is a single bladed powered setup and therefore the local power changes can not be compared with the ones of a standard wind turbine design. Text has been added to clarify this.

**RC2_11:**

[line 55] again modeling details are mixed up with the design aspects.

**AC2_11:**

See RC2_6

**RC2_12:**

[figure 2] It is not clear what are the quantities presented in the plot.

**AC2_12:**

The caption has been updated.

**RC2_13:**

[line 69] % of tip radious?

**AC2_13:**

The text has been updated to include % of tip radius for the locations.

**RC2_14:**

[figure 7] nice artistic picture!

**AC2_14:**

Thank you. The rainbow appeared the right time for the photo.

**RC2_15:**

[line 100] could you provide though some estimate of its flexibility e.g. maximum deflection when thrust is maximized and some natural frequency over rotational frequency. So as to support the assumption.

**AC2_15:**

The maximum tower deflection is estimated to be within 10 cm, with the ratio of its first natural frequency over the rotational frequency around 1.2. The info has been added in the text.

**RC2_16:**

[line 182] not clear. Most probably means that only pressure loads are available. Otherwise I don't see why can't you obtain a chordwise force due to pressures only.

**AC2_16:**

Correct. The text has been updated.

**RC2_17:**

[line 185] reading this paragraph it remains unclear whether most or all simulations shown in the paper are based on fully turbulent polars. Why do you say that these results are omitted? They are shown in a following section.

**AC2_17:**

All results are based on transitional boundary layer characteristics. Since the state of the boundary layer during operation in the field is unknown, the transitional polars have been selected since this was closer to the results. In reality there is a dynamic blend between the two boundary layer states depending on many factors including inflow turbulence. The text has been updated to clarify this.

**RC2_18:**

[line 208] I'm not sure whether sign convention for pitch angle has been defined. Since usually positive pitch is pitch to feather, perhaps it's should be stated that it is the other way around in this work.

**AC2_18:**

The text has been updated to clarify the pitch angle sign convention.

**RC2_19:**

[line 208] Please explain further. Do you mean that average loads increase due to higher than CLmax lift values dictated by stall delay?

**AC2_19:**

Correct. The text has been updated to discuss this.

**RC2_20:**

[line 220] "indications of stall occurrence" maybe reads better.

**AC2_20:**

The text has been updated.

**RC2_21:**

[line 233] please correct

**AC2_21:**

The text has been updated.

**RC2_22:**

[line 240] are also shown here

**AC2_22:**

The text has been updated.

**RC2_23:**

[line 244] delete

**AC2_23:**

The text has been updated.

**RC2_24:**

[line 273] in every

**AC2_24:**

The text has been updated.

**RC2_25:**

[line 317] Very strange behaviour indeed. Maybe deserves some further investigation in the future.

**AC2_25:**

We agree. The need for a future investigation is already mentioned.